# Induction of trained immunity by influenza vaccination - impact on COVID-19

Priya A. Debisarun[1], Katharina L. Gössling[2], Ozlem Bulut[1], Gizem Kilic[1], Martijn Zoodsma[3,4], Zhaoli Liu[3,4], Marina Oldenburg[2], Nadine Rüchel[2], Bowen Zhang[3,4], Cheng-Jian Xu[1,3,4], Patrick Struycken[5], Valerie A. C. M. Koeken[1,3,4], Jorge Domínguez-Andrés[1], Simone J. C. F. M. Moorlag[1], Esther Taks[1], Philipp N. Ostermann[6], Lisa Müller[6], Heiner Schaal[6], Ortwin Adams[6], Arndt Borkhardt[2], Jaap ten Oever[1], Reinout van Crevel[1], Yang Li[3,4], Mihai G. Netea[1,7,8]*

1 Department of Internal Medicine, Radboud University Medical Center, Nijmegen, Netherlands, 2 Department for Pediatric Oncology, Hematology and Clinical Immunology, University Hospital Duesseldorf, Medical Faculty, Heinrich Heine University Duesseldorf, Dusseldorf, Germany, 3 Department of Computational Biology for Individualised Infection Medicine, Centre for Individualised Infection Medicine (CiiM), a joint venture between the Helmholtz-Centre for Infection Research (HZI) and the Hannover Medical School (MHH), Hannover, Germany, 4 TWINCORE, a joint venture between the Helmholtz-Centre for Infection Research (HZI) and the Hannover Medical School (MHH), Hannover, Germany, 5 Department of Occupational Health & Safety, and Environmental Service, Radboud University Medical Center, Nijmegen, Netherlands, 6 Institute of Virology, University Hospital Duesseldorf, Medical Faculty, Heinrich Heine University Duesseldorf, Dusseldorf, Germany, 7 Human Genomics Laboratory, Craiova University of Medicine and Pharmacy, Craiova, Romania, 8 Department for Immunology & Metabolism, Life and Medical Sciences Institute (LIMES), University of Bonn, Bonn, Germany

☯ These authors contributed equally to this work.
* mihai.netea@radboudumc.nl

**Data Availability Statement:** Single-cell sequencing data have been deposited at the European Genome-phenome Archive (EGA), under the accession number EGAS00001005446. Access link: wwwdev.ebi.ac.uk/ega/studies/

## Abstract

Non-specific protective effects of certain vaccines have been reported, and long-term boosting of innate immunity, termed *trained immunity*, has been proposed as one of the mechanisms mediating these effects. Several epidemiological studies suggested cross-protection between influenza vaccination and COVID-19. In a large academic Dutch hospital, we found that SARS-CoV-2 infection was less common among employees who had received a previous influenza vaccination: relative risk reductions of 37% and 49% were observed following influenza vaccination during the first and second COVID-19 waves, respectively. The quadrivalent inactivated influenza vaccine induced a trained immunity program that boosted innate immune responses against various viral stimuli and fine-tuned the anti-SARS-CoV-2 response, which may result in better protection against COVID-19. Influenza vaccination led to transcriptional reprogramming of monocytes and reduced systemic inflammation. These epidemiological and immunological data argue for potential benefits of influenza vaccination against COVID-19, and future randomized trials are warranted to test this possibility.

## Author summary

COVID-19, caused by the virus SARS-CoV-2, has claimed millions of lives and affected many more since its emergence. A number of studies have previously suggested that

EGAS00001005446 The Olink proteomics data have been deposited to the Immport database. Our proteomics data can be accessed via this link: https://www.immport.org/shared/study/SDY1799 or with the DOI 10.21430/M3LIQ1CWAN.

**Funding:** MGN was supported by a Spinoza Grant of the Netherlands Association for Scientific Research (https://www.nwo.nl/en/spinoza-prize) and a European Research Council Advanced Grant (#833247, https://erc.europa.eu/funding/advanced-grants). YL was supported by a European Research Council Starting Grant (#948207, https://erc.europa.eu/funding/starting-grants) and a Radboud University Medical Centre Hypatia Grant (2018, https://www.radboudumc.nl/en/research/academic-and-scientific-training/talent-management/talent-programs/radboudumc-hypatia-track-and-grants). PNO, LM and HS were supported by the Jürgen Manchot Foundation (http://www.manchot.org/). KLG was supported by the German Research Foundation (DFG # 428917761, https://www.dfg.de/en/). MO was supported by the Research Committee of the Heinrich Heine University Duesseldorf (https://www.medizin.hhu.de/en/research/funding/funding-lines-of-the-research-committee). The funders had no role in study design, data collection and analysis, decision to publish, or preparation of the manuscript.

**Competing interests:** The authors have declared that no competing interests exist.

influenza vaccination can provide protection against COVID-19. Although multiple COVID-19 vaccines are currently available, emergence of new variants and inequity in vaccine distribution around the world make it crucial to identify alternative ways that can help in the fight against this pandemic. With this in mind, we investigated if seasonal influenza vaccination had any effect on COVID-19 incidence. Dutch healthcare workers who received the influenza vaccine in the previous flu season had 37% and 49% lower risk of SARS-CoV-2 infection in the first two waves of the pandemic, respectively. We also explored the mechanisms underlying this observation, using various techniques such as single-cell RNA sequencing, proteomics, and stimulation assays. The influenza vaccine reduced systemic inflammation and reprogrammed the immune cells to fine-tune the response against SARS-CoV-2. This study reveals influenza vaccination as a safe and helpful tool to decrease COVID-19 burden.

## 1. Introduction

As of May 2021, over 150 million cases and 3.1 million deaths due to the novel coronavirus disease COVID-19 have been reported [1]. COVID-19 is caused by severe acute respiratory syndrome coronavirus 2 (SARS-CoV-2). While in the majority of cases the virus causes mild symptoms that resolve spontaneously, in the elderly or patients with underlying co-morbidities, the disease is often severe and potentially lethal [2]. Due to the rapid spread and high clinical and socio-economic burden of COVID-19, sustained efforts have been made to develop preventive and therapeutic strategies [3,4]. Several effective anti-COVID-19 vaccines have been designed and successfully tested, with almost 1 billion vaccine doses already administered [1]. However, vaccine supply is still not able to ensure the global needs, with many countries facing challenges in ensuring access to enough COVID-19 vaccines [5]. An additional challenge is the emergence of new SARS-CoV-2 variants which are on the one hand more infective, and on the other hand can display vaccine escape [6].

Due to the absence of specific COVID-19 vaccines in the beginning of the pandemic, as well as the current challenges posed by limited vaccine supply and emergence of new virus strains, vaccination strategies using already available vaccines that can protect against a broad array of pathogens has been proposed as 'bridge vaccination' [7]. The potential interaction between vaccines and infections other than their target disease has attracted a lot of attention lately. It has been demonstrated that certain vaccines, such as Bacillus Calmette-Guérin (BCG), measles-containing vaccines, or oral polio vaccine, have strong non-specific protective effects through long-term reprogramming of innate immunity, a process called *trained immunity* [8]. In line with this, several recent studies suggested a potential link between influenza vaccination and decreased COVID-19 incidence and severity [9–12]. This suggests that influenza vaccination may potentially convey partial protection against COVID-19, and this potential beneficial effect needs to be investigated.

In this study, we assessed the association between influenza vaccination and COVID-19 incidence during the first two waves of the pandemic in the Netherlands, among employees of the Radboud University Medical Center (Radboudumc), a large academic hospital. In addition, as possible mechanisms of action, we investigated the induction of trained immunity and the impact on systemic inflammation by the influenza vaccine used in the 2020 autumn season in 28 healthy adult volunteers.

## 2. Results

### 2.1. Quadrivalent inactivated influenza vaccination is associated with lower COVID-19 incidence

To investigate the effect of influenza vaccination on COVID-19 incidence, we compared the incidence of COVID-19 cases, validated by SARS-CoV-2 PCR, among hospital workers who were either vaccinated or not against influenza.

As of June 1st, 2020, at the end of the first COVID-19 wave in the Netherlands, Radboudumc had 6856 employees working in the clinical departments with direct patient contact (Table 1). The total influenza vaccine coverage rate (VCR) in the hospital for that season (autumn 2019) was 53% (3655/6856). Among these, 184 were documented to have contracted SARS-CoV-2 PCR-positive COVID-19. 42% of SARS-CoV-2 positive individuals during the first wave (77/184) had received an influenza vaccination in the preceding flu season, as opposed to 54% (3578/6672) of SARS-CoV-2 negative employees: 3.34% of the individuals who were not vaccinated against influenza had COVID-19, compared to 2.11% of the vaccinated employees (RR = 0.63, 95% CI, 0.47–0.84, $P$ = 0.0016).

A lower incidence of SARS-CoV-2 positivity among vaccinated individuals was also reported during the second wave (Table 1), when data for the total number of 10899 Radboudumc employees became available. The hospital's total influenza vaccination coverage rate during the 2020/2021 influenza season was 42% (4529/10899). 91 of the 341 SARS-CoV-2 positive employees (27%) were vaccinated against influenza in the autumn 2020. The COVID-19 incidence during the second wave was 3.92% among unvaccinated employees and 2.00% for vaccinated employees (RR of 0.51, 95% CI 0.40–0.65, $P$ < 0.0001).

Our results indicate that influenza vaccination was significantly associated with lower COVID-19 incidence among hospital employees during the first two waves of the pandemic.

### 2.2. Influenza vaccination induces long-term transcriptional reprogramming

To assess a possible induction of trained immunity upon influenza vaccination as the underlying mechanism of protection against SARS-CoV-2, a proof-of-principle study to assess the non-specific immunological effects of the influenza vaccination was performed in 28 healthy individuals. Participants received an influenza vaccine (Influvac Tetra), and blood was collected 1 week before and 6 weeks after vaccination. The study design is summarized in Fig 1.

Long-term transcriptional reprogramming of innate immune cells is a hallmark of the induction of trained immunity. We performed single-cell RNA-sequencing (scRNA-seq) on the peripheral blood mononuclear cells (PBMCs) collected from 10 individuals before and after influenza vaccination. In total, 10,785 cells before and 14,777 cells after vaccination were collected and clustered into 10 subsets, which were annotated by their marker gene expression

**Table 1. COVID-19 incidence among influenza vaccinated and unvaccinated employees of Radboud University Medical Center in the first two waves of the pandemic.** First wave: March—June 2020, second wave: November 2020—January 2021. Influenza vaccinations in autumn of 2019 and autumn of 2020 were considered for calculations regarding the first and the second COVID-19 waves, respectively.

| | | First wave | | | | Second wave | | | |
|---|---|---|---|---|---|---|---|---|---|
| | | SARS-CoV-2 negative | SARS- CoV-2 positive | Total | SARS-CoV-2 incidence | SARS-CoV-2 negative | SARS-CoV-2 positive | Total | SARS-CoV-2 incidence |
| Influenza vaccination | No | 3094 | 107 | 3201 | **3.34%** | 6120 | 250 | 6370 | **3.92%** |
| | Yes | 3578 | 77 | 3655 | **2.11%** | 4438 | 91 | 4529 | **2.00%** |
| | Total | 6672 | 184 | 6856 | **2.68%** | 10558 | 341 | 10899 | **3.13%** |

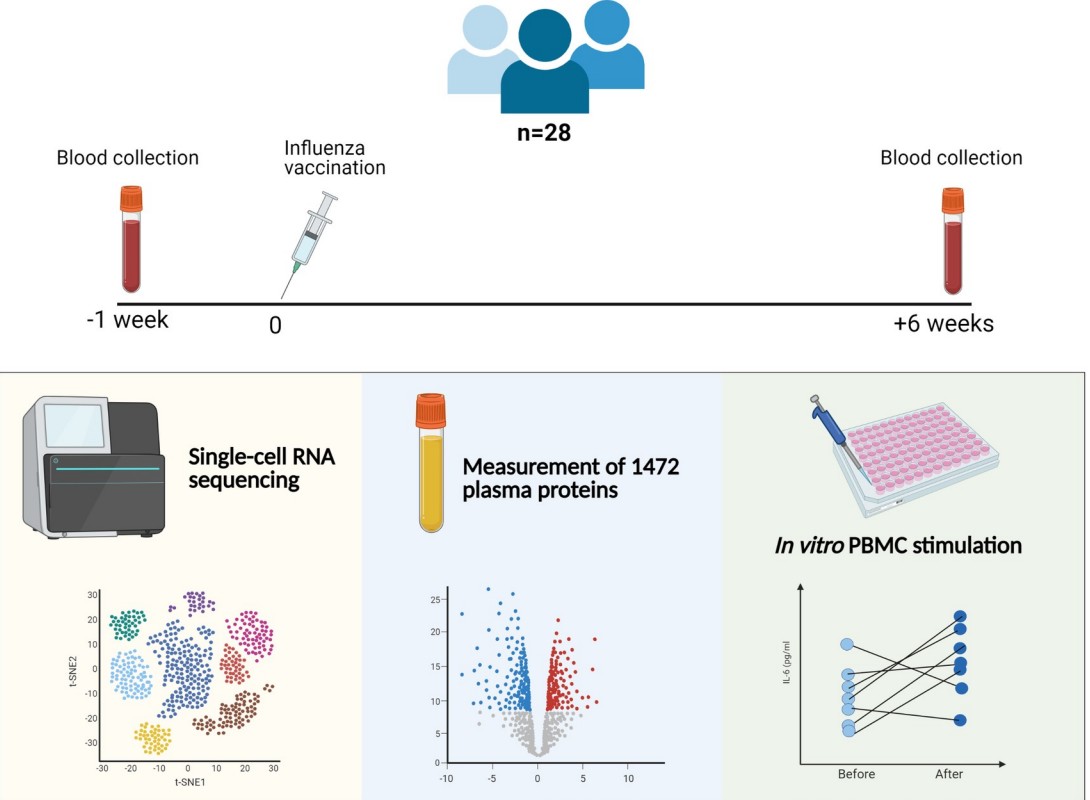

**Fig 1. Summary of the clinical study investigating the effects of an influenza vaccine on trained immunity.** Blood was collected 1 week before and 6 weeks after the influenza vaccination from 28 participants. PBMC stimulation, quantification of plasma proteins, and single-cell RNA sequencing were performed.

as CD14+ monocytes, CD16+ monocytes, CD4+ naïve T cells, memory CD4+ T cells, CD8+ T cells, natural killer cells, B cells, dendritic cells and megakaryocytes (S1A Fig). Immune cell counts were similar after vaccination compared to before (S1B Fig).

While vaccination induced transcriptomic changes in both lymphoid and myeloid cells, some of the most prominent transcriptional changes were observed in CD14+ monocytes (Fig 2A). 136 genes were differentially expressed in CD14+ monocytes, of which 103 were upregulated and 33 were downregulated, comparable with the changes observed in CD4+ naïve T cells (138 differentially expressed genes (DEGs), 105 of them upregulated and 23 downregulated) (Fig 2A and 2C). Among the most differentially expressed genes, myeloid cell nuclear differentiation antigen (*MNDA*) and cathepsin S (*CTSS*) were strongly upregulated in CD14 + monocytes after influenza vaccination (Fig 2A and 2B). Interestingly, three long non-coding RNAs (lncRNAs, the nuclear paraspeckle assembly transcript 1 (*NEAT1*), metastasis associated lung adenocarcinoma transcript 1 (*MALAT1* or *NEAT2*), and splicing factor proline and glutamine rich *SFPQ*), and genes related to the NFKβ signaling pathway (NFKB Inhibitor Alpha (*NFKBIA*), and *JUN*), were downregulated in CD14+ monocytes after vaccination (Fig 2A). Unlike the downregulation in monocytes, *JUN* was upregulated in CD4+ T cells after the vaccination (Fig 2C).

Pathway enrichment analyses revealed that several pathways important for host defense against COVID-19 were upregulated by influenza vaccination: COVID-19 pathway, antigen processing and presentation pathway, while mRNA splicing, histone H3 deacetylation and IL-

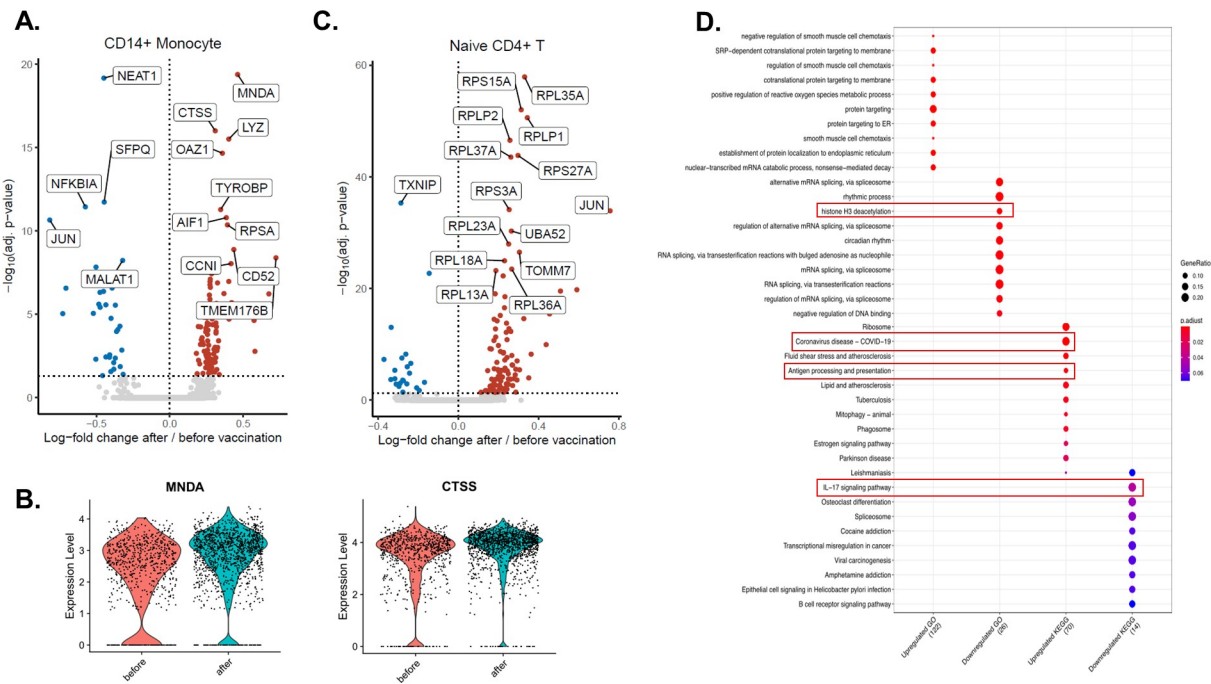

**Fig 2. Transcriptomic changes induced by influenza vaccination on a single cell level. A.** Volcano plot depicting the upregulated and downregulated genes in CD14+ monocytes 6 weeks after influenza vaccination. Red dots show upregulation, while blue dots show downregulation. **B.** Violin plots showing the expression levels of MNDA and CTSS before and after vaccination in CD14+ monocytes. **C.** Volcano plot depicting the upregulated and downregulated genes in CD4+ naïve T cells 6 weeks after influenza vaccination. Significantly changing top 15 genes were labeled on the volcano plots. **D.** Pathway analysis of genes up-regulated or down-regulated by influenza vaccination.

17 pathways were downregulated in CD14+ monocytes (Fig 2D). On the other hand, naïve CD4+ T cells exhibited upregulation in translation, protein localization, and viral gene expression and downregulation in lymphocyte differentiation and NFκB signaling (S2 Fig). Lastly, type I IFN signaling and antigen presentation pathways were among the upregulated pathways in memory CD4+ T cells, while mRNA splicing was downregulated (S3 Fig).

Together, the data show that influenza vaccination induces extensive transcriptomic changes in immune cells, including CD14+ monocytes.

## 2.3. Influenza vaccination downregulates systemic inflammation

Dysregulated inflammatory responses play an important role in the pathogenesis of COVID-19, and we wanted to assess whether influenza vaccination impacts systemic inflammatory reaction. 1472 proteins in the plasma of 10 participants were measured with the Olink proteomics platform before and after vaccination. 368 of the proteins belonged to a panel of inflammatory biomarkers. Principal component analysis performed with these 368 proteins revealed a significant difference between samples before and after influenza vaccination, with a vast majority of inflammatory biomarkers being strongly down-regulated after vaccination (Fig 3A). TNF receptor superfamily member 14 (TNFRSF14), interleukin-1 receptor-associated kinase 1 (IRAK1), and mitogen-activated protein kinase kinase 6 (MAP2K6) were among the proteins that contributed the most to that difference (S4 Fig). Wilcoxon paired signed-rank test identified one upregulated and 82 downregulated proteins in the inflammation panel after influenza vaccination ($P<0.05$; Fig 3B). The only upregulated protein, polypeptide N-acetylgalactosaminyltransferase 3 (GALNT3), is involved in protein glycosylation and important for

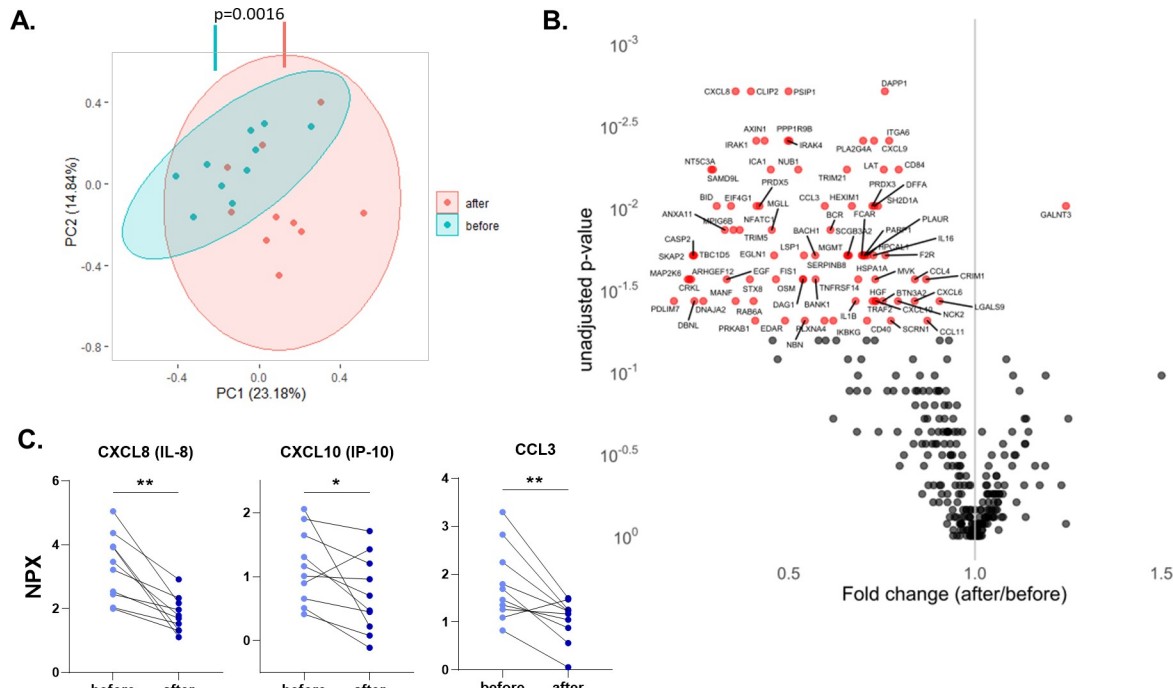

**Fig 3. Influenza vaccination downregulates circulating inflammatory proteins. A.** Principal component analysis (PCA) of circulating proteins belonging to 10 participants before and after vaccination. The mean difference between before and after in PC1 was calculated by Wilcoxon signed-rank test. **B.** Volcano plot depicting the fold changes after influenza vaccination. Red dots show significantly changing proteins. **C.** Selected chemokines whose abundances significantly decrease after the vaccination. Differences between protein expressions before and after influenza vaccination were analyzed using the Wilcoxon signed-rank test, n = 10. NPX: Normalized protein expression. $^*$p<0.05, $^{**}$p<0.01.

oligosaccharide biosynthesis. The 82 downregulated proteins include many chemokines (Fig 3B and 3C), proteins related to tumor necrosis factor (TNF) signaling such as TNF receptor-associated factor 4 (TRAF4) and TNFRSF14, interleukin 1β (IL-1β), IRAK1, IRAK4, and MAP2K6.

Additionally, pathway enrichment analysis was performed with all 283 downregulated proteins, also including proteins from oncology, neurology and cardiometabolic Olink panels (S5 Fig). Noticeable pathways downregulated by influenza vaccination include apoptotic signaling, myeloid cell activation, leukocyte degranulation, and DNA damage response.

Overall, these data demonstrate that influenza vaccination is associated with lower systemic inflammation and downregulation of several inflammatory and apoptotic pathways.

## 2.4. Influenza vaccination influences cytokine production capacity

Next, we investigated whether the influenza vaccine could modulate cytokine responses after stimulation with specific or heterologous ligands and induce trained immunity. Freshly isolated PBMCs before and after influenza vaccination of the volunteers were stimulated *ex vivo* with heat-inactivated SARS-CoV-2, heat-inactivated Influenza H1N1, poly(I:C) (TLR3 ligand), and R848 (TLR7/8 ligand).

Influenza vaccination led to significantly higher TNFα production upon Influenza and poly (I:C) stimulation (Fig 4A). On the other hand, IL-6 production was strikingly lower against SARS-CoV-2 6 six weeks after vaccination (Fig 4B). Similar to TNFα, poly(I:C) induced higher

IL-6 secretion from PBMCs 6 weeks after vaccination, compared to the stimulation one week before the vaccine administration.

We also quantified the two major cytokines of the IL-1 pathway, IL-1β and IL-1Ra, following the exposure of PBMCs with viral stimuli. SARS-CoV-2-induced IL-1β response was significantly reduced after the influenza vaccination, while poly(I:C) resulted in higher IL-1β production (Fig 4C). IL-1β production following SARS-CoV-2 stimulation was below detection limit in 10 individuals before vaccination, and in 15 individuals after influenza vaccination. In contrast, SARS-CoV-2 stimulation induced higher IL-1Ra production from PBMCs 6 weeks after the influenza vaccination, compared to the baseline release before vaccination (Fig 4D). R848 induced similar concentrations of the innate immune cytokines before and after vaccination. However, a significantly higher IL-12 secretion was observed after poly(I:C) and R848 stimulation (Fig 4E). Of note, SARS-CoV-2 and Influenza did not lead to detectable levels of IL-12 in PBMCs.

T cell-derived cytokines, IFNγ, IL-17, and IL-22 produced by PBMCs upon heat-killed Influenza stimulation were similar before and after vaccination (S6 Fig).

Lastly, we investigated whether baseline systemic inflammatory status is associated with the change of anti-SARS-CoV-2 responses upon influenza vaccination. Out of 82 inflammatory mediators downregulated by vaccination, 26 were significantly correlated with the production of at least one cytokine (Fig 5A). An overwhelming majority of them were positively correlated with the increased ratio in the production of the anti-inflammatory cytokines IL-1Ra and IL-10 after vaccination. Inhibitor of NF-κB kinase regulatory subunit gamma (IKBKG) and TRAF2 were positively correlated with the increase in IL-1β induction, while hepatocyte growth factor (HGF) with IL-6 induction (Fig 5A and 5B). Tripartite motif-containing 5 (TRIM5), AXIN1, and galectin 9 (LGALS9) were negatively correlated with the change in TNFα production capacity. Baseline concentrations of GALNT3, the only upregulated protein that belonged to the inflammation panel, were negatively correlated with the induction of anti-SARS-CoV-2 IL-6 response (Fig 5C).

Collectively, the data indicate that seasonal influenza vaccination can induce a more robust innate immune response against viral stimuli such as poly(I:C) and R848. Interestingly, after SARS-CoV-2 stimulation, the trained immunity program induced by influenza vaccination is characterized by lower pro-inflammatory and higher anti-inflammatory cytokine production. Furthermore, higher baseline inflammation was correlated with a more anti-inflammatory response against SARS-CoV-2 after vaccination.

## 3. Discussion

In this study, we present epidemiological and immunological evidence that quadrivalent inactivated influenza vaccination influences the response to and the incidence of SARS-CoV-2 infection. Relative risk reduction for a COVID-19 positive diagnosis in individuals vaccinated against influenza was 37% and 49% during the first and second COVID-19 waves, respectively, suggesting a protective effect of influenza vaccination against infection with SARS-CoV-2. In addition, we found improved responsiveness of immune cells to heterologous viral stimuli after influenza vaccination, arguing for changes in innate immune responses characteristic of a trained immunity program. Moreover, influenza vaccination modulated the responses against SARS-CoV-2, reducing IL-1β and IL-6 production while enhancing IL-1Ra release.

Our observation of reduced incidence of COVID-19 in the employees of a Dutch academic hospital is in line with several recent epidemiological studies (S1 Table). Despite a few reports of a positive association between influenza vaccine coverage and COVID-19 incidence and mortality [13,14], most studies from different countries have revealed a negative association

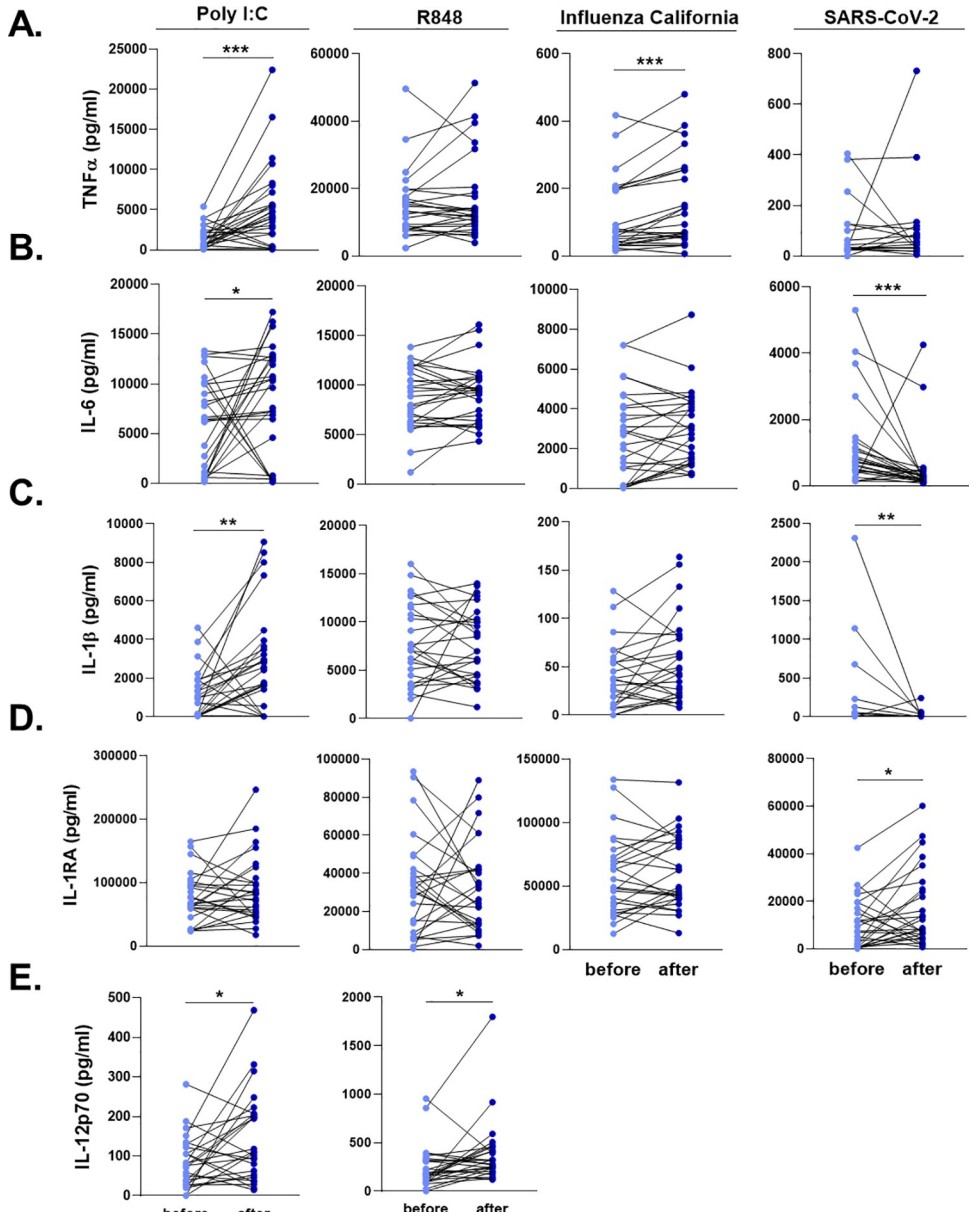

**Fig 4. Ex vivo cytokine responses of the individuals before and after influenza vaccination.** PBMCs were stimulated with heat-killed Influenza H1N1 (California strain), heat-killed SARS-CoV-2, poly I:C, and R848 for 24 hours. **A.** TNFα, **B.** IL-6, **C.** IL-1β, **D.** IL-1RA, and **E.** IL-12 responses were quantified. SARS-CoV-2 and Influenza stimulations did not lead to detectable levels of IL-12. Wilcoxon signed-rank test was performed to compare the differences in cytokine production between before and after influenza vaccination. Light blue dots represent the cytokine values before vaccination, while dark blue dots show after vaccination. n = 28 $^*$p<0.05, $^{**}$p<0.01, $^{****}$p<0,0001.

between influenza vaccination and COVID-19-related hospitalizations, ICU admissions, and mortality [10,15–20]. The strength of epidemiological observations should be interpreted with caution due to the inherent possibility of bias in observational trials. Information on confounding factors and effect modifiers can be missing, and correction for confounders is sometimes not possible, causing over- or underestimation of outcomes. The healthy vaccinee bias could also play a role in the overestimation of the positive effect of a vaccine: individuals willing to be vaccinated against influenza may be also those more likely to respect the personal

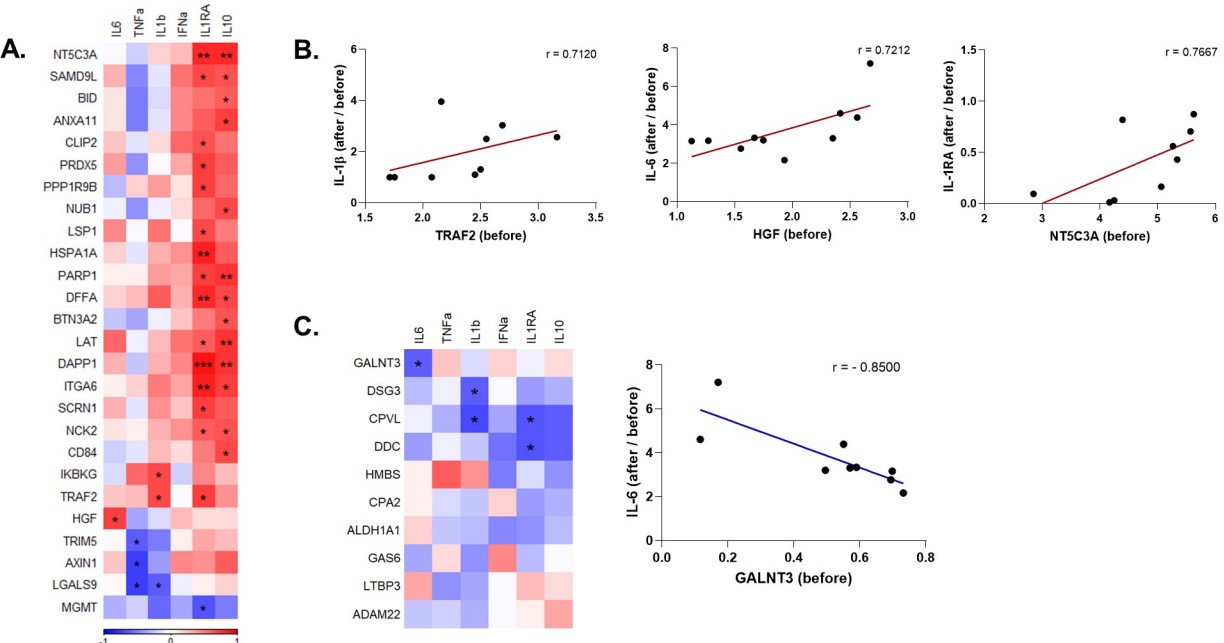

**Fig 5. Correlation of ex vivo anti-SARS-CoV-2 responses with circulating inflammatory mediators that are significantly altered by influenza vaccination. A.** Heatmap depicting the baseline (before vaccination) levels of inflammatory mediators that were downregulated by influenza vaccination and are significantly correlated with the induction of cytokines upon SARS-CoV-2 stimulation before vs. after vaccination. Red indicates positive correlation, and blue indicates negative correlation. **B.** Selected circulating protein and cytokine pairs that are significantly correlated. **C.** Correlation of circulating proteins that were upregulated by influenza vaccination with anti-SARS-CoV-2 cytokine responses, and dot plot depicting the association of baseline GALNT3 levels and induction IL-6 against SARS-CoV-2 after vaccination. r: Spearman's correlation coefficient. *p<0.05, **p<0.01, ***p<0,001.

protection rules against COVID-19 infection. However, an important argument that influenza vaccination could truly exert protective effects against COVID-19 comes from a Brazilian study showing negative correlations between influenza vaccine and COVID-19 mortality, need of ICU treatment, and invasive respiratory support [18]. Correction for comorbidities, several sociodemographic factors, and healthcare facilities uphold the conclusions of this study, and these effects on disease severity in individuals who already contracted COVID-19 cannot be explained by the healthy vaccinee bias.

It has been hypothesized that trained immunity might be an important mechanism underlying these beneficial heterologous effects of vaccines [8]. The most extensively studied vaccine that induces trained immunity is BCG, which is currently being examined for its putative protective effects against COVID-19 duration and severity in several clinical trials (NCT04328441, NCT04348370, NCT04327206, NL8609). Although this property is usually assigned to live vaccines [21], whether inactivated influenza vaccination can also induce trained immunity was not known. Here, we found that Influvac Tetra indeed induces a transcriptional and functional reprogramming of innate immune cells 6 weeks after the vaccination, modulating cytokine responses upon viral challenge with unrelated stimuli.

Single-cell RNA sequencing shows that transcriptional programs of both CD14+ monocytes and CD4+ T cells were considerably altered by the seasonal influenza vaccination. Among the most upregulated genes in monocytes, *MNDA* is an interferon-inducible gene, known to increase in monocytes upon IFNα exposure [22], while Cathepsin S (*CTSS*) participates in the MHC II-dependent presentation of antigens [23]. A similar protein, Cathepsin L, is shown to be critical for SARS-CoV-2 entry through endocytosis [24]. By enhancing the uptake,

processing, and presentation of SARS-CoV-2 antigens, upregulation of *CTSS* might be beneficial to induce the anti-viral immune response. Moreover, *MNDA* is shown to restrict HIV-1 transcription and replication in human macrophages [25], and although not yet explored, similar anti-viral effects may be envisaged against SARS-CoV-2.

Three of the most significantly downregulated genes in monocytes, *NEAT1*, *MALAT1* (NEAT2), and *SFPQ*, contribute to inflammatory responses. Among them, *NEAT1* is associated with anti-viral immunity, IL-8, and type I IFN production [26], while *MALAT1* induces pro-inflammatory cytokine production [27]. In contrast, *SFPQ* blocks *NEAT1* activity by binding to it [26]. Similarly, *JUN* and *NFKBIA*, which are both downregulated in CD14+ monocytes, oppose each other: JUN is a part of AP-1 transcription factor complex stimulating cytokine production [28], while IKBA coded by *NFKBIA* functions to inhibit NFκB signaling, hence inflammatory response [29]. Upregulation of JUN and FOS that form AP-1 together in both CD4+ naïve and memory cells might contribute to improved anti-SARS-CoV-2 T cell response. On the other hand, their downregulation in monocytes might be the underlying mechanism of lower IL-6 and IL-1β production observed after stimulation of immune cells of vaccinated individuals with SARS-CoV-2. These data demonstrates that influenza vaccination induces a fine-tuning of the cytokine production to viral stimulation, that could positively influence a balanced inflammatory reaction during infection.

Previously, our group showed that the BCG vaccine reduces systemic inflammation, and baseline inflammation is linked with trained immunity [30]. In this study, plasma proteomic analyses revealed that influenza vaccination also reduced the level of the systemic inflammation status, similarly to BCG. Only GALNT3 was significantly more abundant in the circulation of vaccinated individuals. GALNT3 is known to be upregulated by Influenza A infection, leading to mucin production and affecting virus replication [31]. Of note, GALNT3 levels were correlated with lower IL-6 response upon SARS-CoV-2 stimulation after influenza vaccination, possibly contributing to the beneficial effects of the vaccine. On the other hand, 82 proteins related to chemotaxis, apoptosis, and TLR signaling pathways were significantly downregulated by vaccination. Caspases CASP3, CASP8, CASP10, and FAS associated death domain (FADD) are important for apoptosis which is a pathway downregulated upon influenza vaccination (S5 Fig). SARS-CoV-2 was shown to induce cell death via activation of caspase-8, which eventually leads to lung damage [32]. Therefore, a lower abundance of the apoptotic proteins in the circulation could contribute to the protection against SARS-CoV-2-related damage.

Systemic low-grade inflammation is associated with poor vaccine and immune responses [33]. This is in line with the results of this study demonstrating that reduced systemic inflammation induced by influenza vaccination is subsequently associated with a higher pro-inflammatory cytokine production against viral stimuli *in vitro*, such as Influenza and poly(I:C). In addition, influenza vaccination was followed by a modified pro/anti-inflammatory balance in the IL-1 pathway, with lower IL-1β and IL-6, but higher IL-1Ra response after SARS-CoV-2 stimulation *in vitro*. Baseline levels of inflammatory modulators were mainly correlated with higher anti-inflammatory cytokine production upon SARS-CoV-2 infection *in vitro*. Fast induction of pro-inflammatory cytokines at the beginning of SARS-CoV-2 infection is crucial to decrease the viral load [34]. However, anti-inflammatory cytokines, such as IL-1Ra, are also necessary to fine-tune the inflammation and counteract excessive inflammation, and it was recently reported to protect against respiratory insufficiency in COVID-19 [35]. Together, these cytokines might contribute to keeping a balance in the inflammatory status of the host [36]. On the other hand, dysregulation of cytokine responses as during the so-called cytokine storm, is associated with severe disease outcomes [37]. Our results indicate that the reprogramming induced by the influenza vaccine could prevent excess inflammation against SARS-CoV-

2. We hypothesize that these transcriptomic and proteomic changes induced by influenza vaccination drive immune cells to a distinct functional reprogramming, leading to a balanced response against SARS-CoV-2.

A recent study investigated the epigenetic and transcriptional reprogramming as well as cytokine responses of immune cells after a trivalent seasonal influenza vaccine (TIV) and AS03-adjuvanted H5N1 influenza vaccine [38]. Immunization with TIV led to repressive epigenomic state in myeloid cells. 30 days after TIV vaccination, PBMCs produced significantly less TNFα, IL-1β, IL-12, IP-10, and IL-1RA upon bacterial and viral challenge, in contrast to our observation of enhanced anti-viral response 6 weeks after immunization with a quadrivalent influenza vaccine. Both TIV and the AS03-adjuvanted vaccine reduced the expression of AP-1 transcription factors including *FOS* and *JUN* which are highly upregulated in our dataset. On the other hand, the adjuvanted influenza vaccine enhanced the accessibility of anti-viral genes and increased the resistance of PBMCs against Dengue and Zika virus infections. This is another important finding suggesting that heterologous protection can be induced by influenza vaccination. However, it is clear that different types of influenza vaccines, also depending on the adjuvants, induce distinct trained immunity programs.

Our study also has important limitations. The hospital population database analysis performed in this study did not allow correction for confounders, as we were not able to access individual characteristics due to hospital privacy policies. A critical possible confounder could be direct patient contact within influenza vaccinated and unvaccinated personnel. However, earlier studies have reported that most SARS-CoV-2 infections in hospital personnel occur in society rather than through patient contact in the hospitals [39–41]. Furthermore, there was no information on comorbidities or other exposures outside the hospital environment. While comorbidities are an important factor related to COVID-19 susceptibility and severity, there are no reasons to expect an unequal distribution of comorbidities among influenza vaccinated and unvaccinated personnel for it to cause skewing of the results. Lastly, one cannot rule out healthy-vaccinee bias, which might lead healthier people to better adhere to annual influenza vaccine recommendations. In addition, the *in vivo* trained immunity effect by influenza vaccination had not been studied as part of a placebo-controlled clinical trial, since healthy volunteers, who had decided autonomously to get vaccinated, were recruited.

In conclusion, we provide observational data suggesting a negative association between the quadrivalent inactivated influenza vaccine and COVID-19 incidence. Additionally, we report first insights into the immunological mechanisms underlying these observations. We show that seasonal influenza vaccination can induce a distinct trained immunity program by reducing systemic inflammation and regulating the transcriptional program and cytokine production of circulating immune cells. Considering these data, influenza vaccination may contribute not only to a reduction of influenza but also to the COVID-19-related burden on the healthcare system. Our data show that influenza vaccination is safe in relation to a later SARS-CoV-2 infection, and phase III clinical trials to assess its effects on COVID-19 are warranted.

## 4. Methods

### 4.1. Ethics statement

For the observational study, ethical waiver was obtained from the Arnhem-Nijmegen Ethical Committee. For the ex-vivo immunological study with human participants, ethical approval was obtained from the Duesseldorf Ethical Committee (study ID 2018_199, amendment 2018–199_1- (5/2020), and amendment 2018_199_2 (12/2020)). Written informed consent was obtained from all participants prior to sample collection. All experiments were conducted in accordance with the Declaration of Helsinki. No adverse events were recorded.

## 4.2. Observational study in healthcare workers

**Study subjects.**   The Radboudumc databases on influenza vaccination status and COVID-19 incidence among employees were provided by the Department of Occupational Health and Safety. The hospital databases of SARS-CoV-2 PCR-positive healthcare workers during the first (March—June 2020) and the second (November 2020—January 2021) COVID-19 waves were consulted, and the corresponding influenza vaccination status of employees was retrieved.

**Data analysis.**   Hospital database analysis was done using GraphPad Prism 8 (CA, USA). To assess the association between COVID-19 incidence and influenza vaccination status, a Chi-square test was used, and the relative risks (RR) are reported. No correction for confounding was possible because no individual characteristics were available from the databases; only influenza vaccination status and COVID-19 history were known. All hospital employees are equally offered an influenza vaccination every year. Vaxigrip Tetra and Influvac Tetra were used during the influenza seasons of 2019/2020 and 2020/2021, respectively. It is important to note that SARS-CoV-2 testing at the beginning of the pandemic was only available for employees who were indispensable for patient care due to a shortage of testing materials.

## 4.3. Ex-vivo study of immune responses following influenza vaccination

**Subjects and study design.**   28 healthy volunteers, employees of the University Hospital Duesseldorf, were enrolled in the study. The average age and BMI were 34.9±8.9 and 22.8±2.8, respectively. 61% of the study participants were female. Participants were vaccinated with 0.5 mL of Influvac Tetra (Abbott Biologicals, IL, USA) intramuscularly. Blood was collected by venous blood puncture 1 week before and 6 weeks after vaccination.

**PBMC isolation and stimulation.**   Venous blood was drawn in 3 mL EDTA tubes. The blood was diluted 1:1 with Phosphate Buffered Saline (PBS). Subsequently, PBMCs were isolated using Ficoll-paque (Sigma Aldrich, Taufkirchen, Germany) density gradient centrifugation. The PBMCs layer was collected and washed twice in cold PBS. Cells were reconstituted in RPMI+, consisting of RPMI-1640 culture medium (Sigma Aldrich, Taufkirchen, Germany) supplemented with 10 µg/mL gentamicin, 10 mM L-glutamine and 10 mM pyruvate (Gibco).

Stimulations were performed in the presence of 2% human AB serum (Sigma Aldrich, Taufkirchen, Germany) for the 24h-stimulations and 10% human serum for the 7-day-stimulations. Cells were incubated at 37˚C with 5% $CO_2$, after 24 hours or 7 days, respectively, supernatants were collected and stored at -80˚C.

**End-concentration of the stimuli: Pathogens and recombinant PAMPs.**   Heat-inactivated Influenza B Brisbane (7,4*10x3 K/mL TCID50), heat-inactivated Influenza H1N1 California (3,24 x 10x5 K/mL TCID50), and heat-inactivated SARS-CoV-2 Wuhan-Hu-1 variant (1,4 x 10x3K/mL TCID50) were used in the study. Viruses had been heat-inactivated for 30 minutes at 60˚C. Heat-inactivation had been checked for sufficiency by cell culture inoculation and subsequent qPCR testing. R848 (Resiquimod, TLR7/8 ligand) (Invivogen, San Diego, CA, USA) (3 µg/mL); Poly I:C (TLR3 ligand) (Invivogen, San Diego, CA, USA) (10 µg/mL).

**Cytokine measurements.**   IL-1β, IL-1Ra, IL-10, IFNα, and IL-12p70 were measured in the cell culture supernatants after 24h stimulation using a custom-made multiplex ELISA kit (Procartaplex, Life Technologies GmbH, Darmstadt, Germany) according to the instructions supplied by the manufacturer. IL-6 and TNFα were measured after 24h stimulation and IL-17, IFNγ and IL-22 were measured after 7 days stimulation in the cell culture supernatant using a duo-set ELISA according to the protocol supplied by the manufacturer (R&D systems, Minneapolis. MN, USA). Differences between cytokine productions before and after influenza

vaccination were analyzed using the Wilcoxon signed-rank test. All calculations were performed in GraphPad Prism 8. A p-value lower than 0.05 was considered statistically significant.

**Proteomics measurements and analysis.** Plasma proteins were measured using the Olink Explore Cardiometabolic, Inflammation, Neurology, and Oncology panels by Olink Proteomics (Uppsala, Sweden). Out of 1472 proteins, 183 had a missing data frequency of 25% and were removed from the analyses. Measurements are denoted as normalized protein expression (NPX) values, which provide relative quantification on a log2 scale. Principal component analysis (PCA), differential expression analysis, and correlations using Spearman's rank-order correlation were performed in R (version 4.0.3) and R Studio (version 1.3.1093). R package *limma* was used for differential expression analysis and p-values $< 0.05$ after Benjamini-Hochberg adjustment were considered significant.

**RNA-sequencing.** Cryopreserved PBMCs from 10 individuals before and after vaccination were used to perform single cell RNA sequencing. Frozen cells were thawed at 37˚C and counted using an automated cell counter. Equal number of cells (3,300 per individual) from 4 different individuals were pooled together and then loaded into the Chromium Controller to separate single cells into Gel Beads-in-emulsion (GEMs). Gene expression libraries were constructed following the standard 10X genomics guides (Chromium Next GEM Single Cell 3' Reagent Kits v3.1 (Dual Index) User Guide, Rev A). Library quality per pool was checked using the Agilent Bioanalyzer High Sensitivity DNA kit. For each pooled library, single-cell RNA sequencing was performed in paired-end mode on NovaSeq 6000 (Illumina) with a depth of 50,000 reads per cell. DNA was isolated from PBMCs and then used for genotyping by Illumina GSA Beadchip.

**Single-cell RNA-sequencing analysis.** Reads from scRNA-seq data were aligned to GRCh38 genome using *cellranger* (v.4.0.0, 10X Genomics) to generate a count matrix for each pool recording the number of transcripts (UMIs) for each gene in each cell. Next, we demultiplexed and clustered cells within each pool into four samples using *souporcell*, a genotype-free method [42]. Then, we matched SNPs called from each demutiplexed-sample to individual genotypes and labeled cells accordingly.

After demultiplexing, we used the R package *Seurat* (v4.0.1) for downstream analysis. For quality control, we excluded cells that met the one of the following criteria: mitochondrial content $> 25\%$, either $< 100$ or $> 3000$ detected genes per cell, and $> 5000$ detected transcripts per cell. Furthermore, mitochondrial genes and ribosomal genes were removed from further analyses. Then, gene expression values were normalized by total UMI counts per cell, multiplied by 10,000 (TP10k) and log-transformed by log10(TP10k+1).

Subsequently, we followed a typical *Seurat* clustering workflow with the following steps: First, we selected the 2,000 most variable features and scaled against the number of UMIs. PCA was performed, followed by Shared Nearest Neighbor (SNN) Graph construction using PC1 through PC20 to identify cell clusters. Finally, Uniform Manifold Approximation and Projection (UMAP) was used to visualize the cell clusters. Cell type annotation was based on the following canonical gene markers and combined with SingleR [43] unsupervised annotation results: Naïve CD4+ T (IL7R, TCF7, CCR7), Memory CD4+ T (IL7R), CD8+ T (CD8A, CD8B, NKG7), NK (GZMB, NKG7, GNLY), CD14+ Monocytes (CD14, LYZ), CD16+ Monocytes (FCGR3A), B (CD79A, MS4A1), DC (IRF8, TCF4), megakaryocytes (PPBP).

**Differential expression and enrichment analyses in scRNAseq.** Differential expression analysis was performed in Seurat using *FindMarkers* using the Wilcoxon rank sum test. Genes were considered differentially expressed if they were expressed in at least 10% in either tested group, and the p-value after Bonferroni post-hoc correction was $< 0.05$. Significantly differentially expressed genes between the conditions were retrieved per cell type and used as input for

GO enrichment using *ClusterProfiler* (v.3.18.1) [44]. Enrichment of genes was tested both in Gene Ontologies (GO) and within the Kyoto Encyclopedia of Genes and Genomes (KEGG) and considered significant if the Benjamini-Hochberg adjusted p-value was < 0.05.

## Supporting information

**S1 Fig.** A. Two-dimensional Uniform Manifold Approximation and Projection (UMAP) embedding of 25.562 single cells. Cells are colored respective to their major cell lineages. B. Proportions of immune cell populations 1 week before and 6 weeks after the influenza vaccination.
(TIF)

**S2 Fig. Gene ontology (GO) and Kyoto Encyclopedia of Genes and Genomes (KEGG) pathway enrichment analyses.** Analyses were performed with the genes whose expressions significantly change after vaccination in CD4+ naïve T cells.
(TIF)

**S3 Fig. Gene ontology (GO) and Kyoto Encyclopedia of Genes and Genomes (KEGG) pathway enrichment analyses.** Analyses were performed with the genes whose expressions significantly change after vaccination in CD4+ memory T cells.
(TIF)

**S4 Fig. Contributions of individual proteins to the first two dimensions of the PCA performed with proteomics data before and after vaccination.** Left panel shows the proteins contributing to the variance in the first dimension (PC1), while right panel demonstrates the proteins contributing to the variance in the second dimension (PC2).
(TIF)

**S5 Fig. GO and KEGG pathway enrichment analyses using the proteins whose plasma concentrations significantly change after vaccination.** The complete Olink panel of 1472 proteins were used as the background in the enrichment analysis.
(TIF)

**S6 Fig. Ex vivo IFNγ, IL-17, and IL-22 responses of the individuals before and after influenza vaccination.** PBMCs were stimulated with heat-killed Influenza H1N1 (California strain) for 7 days. IFNγ, IL-17 and IL-22 responses were quantified. Wilcoxon signed-rank test revealed no significant differences in cytokine production between before and after influenza vaccination.
(TIF)

**S1 Table. Studies on the association between influenza vaccination and COVID-19 related outcomes.**
(DOCX)

## Acknowledgments

We thank the Department of Occupational Health and Safety of Radboud University Medical Center, Nijmegen for providing the Influvac Tetra vaccine.

## Author Contributions

**Conceptualization:** Priya A. Debisarun, Katharina L. Gössling, Ozlem Bulut, Gizem Kilic, Patrick Struycken, Heiner Schaal, Ortwin Adams, Arndt Borkhardt, Jaap ten Oever, Reinout van Crevel, Yang Li, Mihai G. Netea.

**Data curation:** Priya A. Debisarun, Katharina L. Gössling, Ozlem Bulut, Gizem Kilic, Patrick Struycken.

**Formal analysis:** Priya A. Debisarun, Katharina L. Gössling, Ozlem Bulut, Gizem Kilic, Martijn Zoodsma, Zhaoli Liu, Bowen Zhang, Cheng-Jian Xu, Valerie A. C. M. Koeken.

**Funding acquisition:** Katharina L. Gössling, Yang Li, Mihai G. Netea.

**Investigation:** Priya A. Debisarun, Katharina L. Gössling, Marina Oldenburg, Nadine Rüchel, Simone J. C. F. M. Moorlag, Esther Taks.

**Methodology:** Priya A. Debisarun, Katharina L. Gössling, Marina Oldenburg, Nadine Rüchel, Philipp N. Ostermann, Lisa Müller.

**Project administration:** Priya A. Debisarun, Katharina L. Gössling.

**Resources:** Mihai G. Netea.

**Supervision:** Jorge Domínguez-Andrés, Yang Li, Mihai G. Netea.

**Visualization:** Priya A. Debisarun, Ozlem Bulut, Gizem Kilic, Martijn Zoodsma.

**Writing – original draft:** Priya A. Debisarun, Ozlem Bulut, Gizem Kilic.

**Writing – review & editing:** Katharina L. Gössling, Martijn Zoodsma, Zhaoli Liu, Marina Oldenburg, Nadine Rüchel, Bowen Zhang, Cheng-Jian Xu, Patrick Struycken, Valerie A. C. M. Koeken, Jorge Domínguez-Andrés, Simone J. C. F. M. Moorlag, Esther Taks, Philipp N. Ostermann, Lisa Müller, Heiner Schaal, Ortwin Adams, Arndt Borkhardt, Jaap ten Oever, Reinout van Crevel, Yang Li, Mihai G. Netea.

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
