## [Decision Letter · Decision Letter 0]

1 Oct 2021

Dear Dr. Netea,

We are pleased to inform you that your manuscript 'Induction of trained immunity by influenza vaccination - impact on COVID-19' has been provisionally accepted for publication in PLOS Pathogens.

Best regards,

Sabra L. Klein

Associate Editor

PLOS Pathogens

Andrew Pekosz

Section Editor

PLOS Pathogens

Kasturi Haldar

Editor-in-Chief

PLOS Pathogens

orcid.org/0000-0001-5065-158X

Michael Malim

Editor-in-Chief

PLOS Pathogens

orcid.org/0000-0002-7699-2064

The reviewers were very positive about the submitted work and had no major concerns. Thank you for submitting your work to PLoS Pathogens.

Reviewer Comments (if any, and for reference):

Reviewer's Responses to Questions

**Part I - Summary**

Reviewer #1: Debisarun et al. present an interesting and timely paper demonstrating a protective effect of influenza vaccination against COVID-19. The authors reviewed vaccination and COVID-positivity data from a large Dutch teaching hospital during two waves of COVID-19 in 2020 and found that COVID-19 PCR positivity was reduced by about one-third among influenza vaccinated hospital staff compared with unvaccinated staff. The team went on to study 28 healthy volunteers who gave blood samples before and after influenza vaccination. Using scRNAseq, they found that vaccination upregulated the expression of Ag processing and presenting genes in monocytes, and it upregulated Ag presentation and IFN signaling in CD4+ memory T cells. Using cytokine proteomics of 1472 plasma proteins, they also found that influenza vaccination downregulated systemic inflammation. Lastly using ex vivo PBMC stimulation assays, the team describe pre- and post-vaccination changes; most interestingly influenza vaccination reduced IL-6 production upon challenge with heat-killed SCV2.

This work is a valuable extension of a larger body of studies (much of it from this same author group) showing protection by vaccines (influenza vaccine, yellow fever vaccine, and BCG) against heterologous challenges including COVID-19.

Reviewer #2: Debisarun and colleagues make an intriguing case that influenza vaccination induced the trained immune phenotype which increases resistance to SARS-CoV-2 infection. Several reports in the literature have indicated that influenza vaccination may increase resistance to COVID-19. The present study offers a potential mechanistic explanation for the negative association between flu vaccine and COVID-19 incidence. When considered as a whole, this is an important and timely report that has significant clinical and scientific implications.

This is a clear and well-written manuscript. The study is strengthened by a combination of epidemiological data derived from hospital workers and observational data derived from influenza vaccinated healthy subjects. The data indicate that vaccination against influenza induces transcriptional and functional changes in peripheral blood monocytes (and perhaps other leukocytes) which culminate in an anti-inflammatory response to SARS-CoV-2 virions. However, this does not appear to be a broad suppression of immunity because antigen presentation is upregulated. One interpretation is that the flu vaccine has reprogrammed or fine tuned (according to the authors) the innate immune response such that exaggerated pro-inflammatory responses are reduced, thus leading to better outcomes.

As with all studies of this type, there are limitations. However, the authors have done an excellent job of identifying the limitations with the study and they have clearly articulated these limitations in the discussion. Not surprisingly, this study raises more questions than it answers, but overall the strengths of the manuscript significantly outweigh the weaknesses.

**Part II – Major Issues: Key Experiments Required for Acceptance**

Reviewer #1: None

Reviewer #2: None.

**Part III – Minor Issues: Editorial and Data Presentation Modifications**

Reviewer #1: 1. The authors do a nice job of pointing out confounders in the clinical study such as the “healthy vaccinee bias”. It would be helpful to also mention whether they had any evidence for hospital employees receiving influenza vaccines or COVID-19 PCR testing outside of the Radboudmc and if so could these have been disproportionate between the vaccinees and non-vaccinees.

2. Lines 399-403. Was the prior vaccination history of the 28 volunteers available? The interval from the most recent prior influenza vaccination and whether they had ever received BCG or had a BCG scar would be valuable to know.

3. Line 198 and beyond. For the cytokine studies, were all 28 volunteers studied? Please explain why there seem to be lower numbers of subjects in certain panels of Fig 4 such as 4C with heat-killed SARS-CoV-2 stimulation.

4. Line 198 and beyond. For the cytokine studies it appears from Fig. 5A that IL-10 was studied, but the data for IL-10 responses are not shown in Figure 4.

5. Lines 218-227. In several places the authors mention the in vitro stimuli as “SARS-CoV-2” or “influenza”. It would be better state in each instance that it was “heat-killed SARS-CoV-2” and “heat-killed influenza”.

6. Line 522. What journal?

Reviewer #2: None.

PLOS authors have the option to publish the peer review history of their article (what does this mean?). If published, this will include your full peer review and any attached files.

Reviewer #1: No

Reviewer #2: No

---

## [Editor Report · Acceptance letter]

20 Oct 2021

Dear Dr. Netea,

We are delighted to inform you that your manuscript, " Induction of trained immunity by influenza vaccination - impact on COVID-19 ," has been formally accepted for publication in PLOS Pathogens.

Best regards,

Kasturi Haldar

Editor-in-Chief

PLOS Pathogens

orcid.org/0000-0001-5065-158X

Michael Malim

Editor-in-Chief

PLOS Pathogens

orcid.org/0000-0002-7699-2064